# Nitrogen Supply Drives Senescence-Related Seed Storage Protein Expression in Rapeseed Leaves

**DOI:** 10.3390/genes10020072

**Published:** 2019-01-22

**Authors:** Stefan Bieker, Lena Riester, Jasmin Doll, Jürgen Franzaring, Andreas Fangmeier, Ulrike Zentgraf

**Affiliations:** 1Centre of Molecular Biology of Plants, University of Tübingen, Auf der Morgenstelle 32, D-72076 Tübingen, Germany; stefan.bieker@zmbp.uni-tuebingen.de (S.B.); lena.riester@zmbp.uni-tuebingen.de (L.R.); jasmin.doll@zmbp.uni-tuebingen.de (J.D.); 2Institute of Landscape and Plant Ecology, University of Hohenheim, August-von-Hartmann-Str. 3, D-70599 Stuttgart, Germany; Juergen.Franzaring@uni-hohenheim.de (J.F.); andreas.fangmeier@uni-hohenheim.de (A.F.)

**Keywords:** senescence, nitrogen remobilization, nitrogen supply, oil seed rape, seed storage proteins, hydrogen peroxide

## Abstract

In general, yield and fruit quality strongly rely on efficient nutrient remobilization during plant development and senescence. Transcriptome changes associated with senescence in spring oilseed rape grown under optimal nitrogen supply or mild nitrogen deficiency revealed differences in senescence and nutrient mobilization in old lower canopy leaves and younger higher canopy leaves. Having a closer look at this transcriptome analyses, we identified the major classes of seed storage proteins (SSP) to be expressed in vegetative tissue, namely leaf and stem tissue. Expression of SSPs was not only dependent on the nitrogen supply but transcripts appeared to correlate with intracellular H_2_O_2_ contents, which functions as well-known signaling molecule in developmental senescence. The abundance of SSPs in leaf material transiently progressed from the oldest leaves to the youngest. Moreover, stems also exhibited short-term production of SSPs, which hints at an interim storage function. In order to decipher whether hydrogen peroxide also functions as a signaling molecule in nitrogen deficiency-induced senescence, we analyzed hydrogen peroxide contents after complete nitrogen depletion in oilseed rape and Arabidopsis plants. In both cases, hydrogen peroxide contents were lower in nitrogen deficient plants, indicating that at least parts of the developmental senescence program appear to be suppressed under nitrogen deficiency.

## 1. Introduction

Despite being a member of the glucosinolate producing Brassicaceae family, selection and breeding of oilseed rape (OSR, *Brassica napus* L.) has made it one of the most important oilseed crops after soybean and oil palm. In addition to high oil contents, its seeds contain high amounts of protein, constituting up to 20%–25% of seed dry weight [1]. Around 60% of this protein content are composed of seed storage proteins (SSPs), which in turn are mainly comprised of two protein families: 12S globulins (Cruciferins) and 2S albumins (Napins) [2,3]. Cruciferins and napins have distinct molecular characteristics and respond differentially to the changes in pH and temperature, therefore their functionalities are most likely contrasting rather than complementary [4]. SSPs are synthesized in the endoplasmatic reticulum and then stored in protein storage vacuoles. Seed storage reserves accumulate with progression of seed growth and development and in mature oilseed rape seeds, the endosperm and the cells of embryo are packed full of protein storage vacuoles and oil bodies. The protein content in the seeds of the Brassica species is affected by the translocation of amino-N and loading of amino acids in the phloem sap of leaves, indicating that seed filling and seed quality strongly rely on remobilization of previously acquired nutrients from the leaves. Phloem unloading in the endosperm and embryo cells by specific transporters also plays an important role. In Arabidopsis, amino acid permease 1 and 2 (AAP1/2) and the cationic amino acid transporter 6 (AtCAT6) mediate AA uptake in the embryo and have an impact on SSP accumulation [5]. QTL mapping identified a total of 67 and 38 QTLs for seed oil and protein content and provided new insights into the complex genetic mechanism of oil and protein accumulation in the seeds of OSR [6]. During germination, the seed storage reserves deposited in the protein storage vacuoles and oil bodies in the endosperm and embryo tissue are degraded for energy supply and anabolic processes of seedling development. In-depth proteomic dissection contributed to a better understanding of the mobilization of seed storage reserves and regulatory mechanisms of the germination process in *B. napus* [7].

Before anthesis, sequential leaf senescence leads to the repartitioning of nutrients from older leaves to newly developing non-reproductive organs. The bottleneck of weak nitrogen (N) remobilization associated with senescence in vegetative stages appears not to be amino acid transport from leaf to phloem but rather an incomplete hydrolysis of foliar proteins [8]. After anthesis, monocarpic leaf senescence governs the nutrient reallocation to the now developing reproductive organs and therefore, has a critical impact on yield quality and quantity. For example in wheat, leaf-derived N remobilized by senescence processes accounts for up to 90% of the total grain N-content [9]. Consistent with the importance of senescence for the plants’ reproductive success, highly controlled processes are put in place to govern it. A complex interplay between hormone action, genetic reprogramming as well as biotic and abiotic factors administer initiation, progression and termination of senescence and thus influences the outcome of nutrient recycling. In Arabidopsis, differential regulation of more than 6000 genes during onset and progression of leaf senescence emphasizes the importance of this developmental program [10]. In this analysis, autophagy, transport and response to reactive oxygen species (ROS) are the first processes activated in the chronology of leaf senescence [10]. ROS, especially H_2_O_2,_, are well known signaling molecules in senescence and we have demonstrated that developmental senescence in Arabidopsis as well as in OSR is associated with the down-regulation of central components of the anti-oxidative systems and thus associated with a transient increase in intracellular H_2_O_2_ contents [11,12]. If this H_2_O_2_ signal is suppressed in Arabidopsis, senescence is delayed [11].

Even though *B. napus’* performance regarding uptake of inorganic N is relatively high, N-remobilization during leaf senescence is believed to not be very efficient [13]. As a consequence of high demand and uptake during vegetative growth but rather low remobilization afterwards, surrounding ecosystems are often polluted by leaching of remaining soluble leaf-nitrogen (NO_3_^−^) into the water as well as by volatilization of N_2_O and NH_3_ into the atmosphere. In addition to the detrimental impacts on environment and water supply by oversaturation with nitrogenous compounds, production of inorganic nitrogen fertilizers is highly resource- and cost-intensive. Enormous progress has already been made by breeding plants with respective traits for high nutrient uptake and remobilization efficiencies, high photosynthetic rates, high sink capacities, variation of fatty acid contents etc. However, there seems to be a natural trade-off between high yield and high protein content [5,14,15]. Hence, besides new breeds, genetically modified crops seem to be one of the most viable and tangible options, not only to avoid a further increase of environmental pollution, but also to tackle upcoming bottlenecks in the global food supply.

Developmental senescence progresses sequentially from the lower to the upper leaves in which the sink leaves in young plants later turn into source leaves during pod ripening [16]. N-deficiency leads to earlier induction of senescence in older leaves, while senescence is delayed at higher leaf positions [17,18]. Transcriptome analyses of leaves in two canopy levels over development of plants grown under two N-fertilization regimes also revealed opposite effects of N-depletion on senescence in lower versus upper canopy leaves. Several transcriptional regulators and protein degradation genes were identified to be differentially expressed in N-depleted lower leaf positions [19]. Morphology and performance analyses of exactly the same plants used in this study have been previously reported by Franzaring et al. [20] who analyzed the effect of todays and future CO_2_ and its interaction with nitrogen supply. Even though the flowering window of OSR was largely extended due to elevated CO_2_, plants were not able to produce more pods but strongly branched out and produced many side shoots pointing to a prevented apical switch-off by high CO_2_ [20]. N-remobilization was more affected by the different N-supply than by the CO_2_ enrichment. Under ambient CO_2_ concentrations, the nitrogen use efficiency (NUE) of the seeds was reduced by 2%, 33% and 65% under low, optimal, and high N treatments, respectively [21]. Moreover, ^15^N-labeled fertilizer was supplied at three different time points to follow up nitrogen recovery over time and the distribution of the nutrient between source and sink organs. ^15^N-supplied at the beginning of plant development mainly accumulated in roots and shed senescent leaves while ^15^N-supplied at later stages ended up in late leaves, stem and reproductive tissue [21].

In order to understand the nitrogen remobilization processes in OSR under different nitrogen regimes more precisely, we analyzed the transcriptome data, which have been produced by Safavi-Rizi and co-workers [19] for genes involved in nitrogen remobilization and metabolism. Surprisingly, we realized that a remarkable amount of SSPs or proteins related to SSP expression were also expressed in leaf tissue, which we followed up in more detail to get some clues about their function in leaves. In a new growth experiment, we confirmed that the SSP genes *CRUCIFERIN* and *NAPIN* were expressed and that the corresponding proteins are produced in leaf tissue. *CRUCIFERIN* mRNA accumulated in leaf tissue at the onset of leaf senescence under high but not under low N-supply. In contrast, *NAPIN* expression was higher in leaves fertilized with low amounts of N but lower under high N-supply. SSP production progressed from the oldest to the youngest leaves and correlated with increasing hydrogen peroxide levels. SSPs were also detected in stem tissue and pod walls pointing to a possible function as an interim N-storage during N-remobilization. *In-silico* analysis of the corresponding promoter sequences identified motifs suggesting a (redox-) stress and nutritional supply dependent expression control. Moreover, when N was completely withdrawn, N-starvation induced premature senescence, but at the same time, a reduction of intracellular H_2_O_2_ contents in OSR as well as in Arabidopsis was observed. This indicates that N-starvation induced senescence is driven by different signals compared to developmental senescence and that at least the hydrogen peroxide-driven parts of the developmental senescence processes appear to be suppressed in N-starvation induced senescence.

## 2. Material and Methods

### 2.1. Plant Growth Conditions

Oilseed rape plants (*Brassica napus* cv Mozart):

*Nitrogen starvation and developmental OSR series on soil:* Plants were grown in greenhouses at 19–22 °C, with a 16 h photoperiod. All plants were sown on *Einheitserde classic-Topferde* (Einheitserde Werkverband e.V., Sinntal-Altengronau, Germany) in 6 × 6 cm pots. Later, plants for N-starvation experiments were put on a nutrient depleted soil (*Einheitserde Typ 0*). This was combined with repotting at around 4 weeks of plant age. Here, only control plants received fertilization with *Wuxal ^®^ Super* (Wilhelm Haug GmbH & Co KG, Ammerbuch-Pfäffingen, Germany) containing 99.2g N/L. Each pot was watered with 1 L of a 0.2% fertilizer solution once a week. After 4 days of habituation, harvests were conducted as indicated. Plants for developmental senescence assays were kept on *Einheitserde classic-Topferde*. Plants were sown in a weekly rhythm for 16 weeks in the green house. Leaf positions 5, 8, 12, the leaf residing directly below the first developing pods (terminal leaf/T-Leaf), the stem between T-leaf and first pod, and the siliques were sampled on the same day at the same time point.

*Field-near conditions:* Climate conditions were simulated in growth chambers at the University of Hohenheim, using the seasonal changes of day length and temperature of South-Western Germany as described in Franzaring, Weller, Schmid and Fangmeier [20]. N was supplied in three different regimes, namely *low (N_L_)*, *optimal (N_O_)* and *plus (N_P_),* fertilized with 75, 150 and 225 kg ha^−1^, respectively. Nutrient supply and atmospheric conditions were kept as described in Franzaring, Gensheimer, Weller, Schmid and Fangmeier [21]. Plants were ^15^N-labeled with double labeled ammonium nitrate with a ^15^N excess of 10% (δ^15^N of 100‰) at DC 0 (first fertilization at 0 DAS, “old N”), at DC 30 (second fertilization at 72 DAS) or at DC 59 (third fertilization at 80 DAS, “new N”) as described in Franzaring, Gensheimer, Weller, Schmid and Fangmeier [21].

*Nitrogen starvation on hydroponics:* Plants were germinated at a 16 h photoperiod, 18–22 °C and 75% relative humidity (RH) on filter paper sandwiched between sponges and PVC-plates on both sides. After seven days, the seedlings were transferred to pots with 6 liters liquid medium in a greenhouse with also 16 h photoperiod and 75% RH. After 28 days pre-culture at 2 mM N, N-starved plants received low N liquid medium (0.1 mM N), full N plants continued receiving N-rich medium (4 mM N). For more detailed composition see Appendix A (published in Koeslin-Findeklee et al. [22])

*Arabidopsis thaliana* Col-0 plants: Plants were grown on a modified Araponics© system (Araponics SA, Liege, Belgium) including aeration of the liquid medium. Moreover, plants were grown on sterilized mineral-wool instead of the recommended agarose. Seeds were stratified for 4 days at 4 °C, and then moved into growth chambers. Temperature was kept at 21 °C, day length was 8 h. Liquid media were exchanged every second day. During the first week after germination, tap water was used as hydroponic medium. Afterwards, plants received 0.5 mM N during the second week and 1 mM N during the third week. With the beginning of the treatment (the fourth week onwards) 0 and 4 mM were supplied to N-starved and control plants, respectively (for more detailed media composition and schedule see Appendix A).

### 2.2. H_2_O_2_ Measurements

Leaf discs (1 cm diameter, *B. napus*) or whole leaves (*A. thaliana*) were sampled from the same leaf positions and then incubated in MS-Medium (pH 5.7) with 9.5 µM 5(6)-Carboxy-Di-Hydro-Di-Chloro-Fluorescein-Di-Acetate (Carboxy-H_2_DCFDA). After 45 min incubation, samples were rinsed twice with distilled water and frozen in liquid nitrogen. Homogenization was conducted on ice in 500 µL 40 mM Tris-HCl pH 7. After 30 min of centrifugation at 4 °C and 14,000 rpm, fluorescence of supernatant was determined (480 nm excitation, 525 nm emission, Berthold TriStar LB941, BERTHOLD TECHNOLOGIES, Bad Wildbad, Germany). H_2_DCFDA solution was prepared freshly for each harvest and calibrated by chemical de-acetylation and oxidation following Cathcart et al. [23].

### 2.3. Generation of Anti-SSP Antisera

Antisera were generated against VVEFEDDA (NAPIN) and VVRPLLRQ (CRUCIFERIN) peptides. Peptide synthesis and animal immunization were carried out by *BioGenes* (Berlin, Germany). Peptide sequences were chosen based on alignments of Arabidopsis and *B. napus* seed storage protein. Antisera were used directly without further purification.

### 2.4. Protein Extraction and Western-Blotting

Samples were homogenized on liquid nitrogen. Protein extraction buffer was added according to the sample amount (modified QB-Buffer; 100 mM KPO_4_, 1 mM EDTA, 1% Triton-X100, 10% glycerol, 1 mM DTT, 150 mM NaCl), samples were then incubated for 20–30 min on ice. After subsequent centrifugation at 20.800 g and 4 °C for 20–60 min (depended on sample type: shoot material with many fibers as well as oily seed material were centrifuged longer) total protein content of the supernatant was determined. 15 µg total protein of the very same *B. napus* (cv. Mozart) seed sample as positive control and 25 µg of total protein of all other samples were then separated by SDS-DISC-PAGE (15% PAA separating, 3.5% PAA stacking gel). After transfer on PVDF-membranes by semi-dry blotting, membranes were washed twice with TBS-T, incubated for 1 h in 5% milk-powder TBS-T and after another washing with TBS-T incubated in 1.5% milk-powder TBS-T with 1:500 diluted antisera for 1–2 h. Anti-rabbit-HRP antibody (CellSignaling, #7074) was used as secondary antibody in a 1:3,000 dilution in 1.5% milk-powder TBS-T. Blots were incubated another hour in this solution. After three times washing with TBS-T, detection was carried out with luminol detection kit (*Pierce ECL* or *BioRad Clarity*) in an *Amersham Imager 600* (GE Healthcare, Berlin, Germany).

For relative quantification, the gel analyzer plugin of *ImageJ* was used. Integrated band intensities were normalized to bands of the respective positive controls. As different antisera were used, NAPIN and CRUCIFERIN signals could not be directly compared and signal intensities were normalized to appropriate bands in whole seed extracts that served as positive controls.

### 2.5. RNA Extraction and qRT-PCR

Primer design for qRT-PCR was done via QuantPrime [24]. RNA extraction and cDNA synthesis were conducted with InviTrap Spin Universal RNA Mini Kit (Invitek Inc, San Francisco, CA, USA) and qScript cDNA SuperMix (Quanta Biosciences, Beverley, Massachusetts, USA) according to the manufacturer’s protocols. qRT-PCR was performed with Perfect CTa SybrGreen Fast Mix (Quanta Biosciences, Beverley, MA, USA) in an iCycler IQ system (Biorad, München, Germany). Relative quantification to *ACTIN2* was calculated with ΔΔC_T_–Method [25]. Primers used for *B. napus NAPIN2 (Bna.2093)* were: 5’-TGG CAA GCT CTT AGG TGT TGA GC (FW) and 5’-CCG GCC CAT TTA GGA TTC CAA G (REV). For *CRUCIFERIN1 (Bna.2089)*: 5’-AGA CCA CTT TGA CGC ACA GCA G (FW) and 5’-AAG CCC TTA AGC ATC AGC CTT CC (REV).

### 2.6. Chlorophyll Measurements

Chlorophyll (Chl) contents were estimated via a *SPAD 502 (KONICA MINOLTA)* photometer or the *atLEAF + (FT Green, LLC)* device. Measurements were conducted at least three times per leaf on varying positions to avoid positional effects.

### 2.7. Microarray Data Evaluation

The plant material of *Brassica napus* cv. Mozart, which was used for transcriptome analyses, the design of the *B. napus* custom microarray, the identification of *Arabidopsis thaliana* homologues of the *B. napus* unigenes, and the basic data analysis workflow are described in Safavi-Rizi et al. [19]. In this experiment, oilseed rape plants were grown under two different nitrogen regimes, optimal (N_O_) and low (N_L_) nitrogen. The lower canopy leaf No. 4 was harvested at 78, 85, 92, and 99 days after sowing (DAS) and the upper canopy leaf No. 8 at 92 and 106 DAS (Figure 1A). Physiological analysis with respect to carbon and N-dependent effects as well as detailed description of growth conditions were published in Franzaring, Weller, Schmid and Fangmeier [20] and Franzaring, Gensheimer, Weller, Schmid and Fangmeier [21]. Briefly, expression data analysis was conducted in R [26] with the *Linear Model for Microarray and RNA-Seq Data* (LIMMA) package [27] applying the standard time-course experiment workflow. Contrast matrixes were built within one treatment with a *p*-value cut-off 0.05 (Benjamini Hochberg correction) and log2 fold-change cut-off of 1. *B. napus* genes were annotated by their most similar *A. thaliana* orthologue, identified by local BLAST+ (version 2.2.30 +, build Aug 28 2015 11:17:27, [28]) against the *TAIR*-10-Database [29]. Blast results with an e-value > 10^−3^ were excluded from further analysis. Finally, all transcripts associated with SSPs were extracted by matching *AGI* identifiers to a list of identifiers of SSP associated genes (for listings see Appendix A).

### 2.8. Total N and δ^15^N-Measurements

Seeds were ground in liquid nitrogen. Subsequently, the homogenized powder was lyophilized and total N-content was measured by a CN-element analyzer (Elementar Vario EL III, Elementar Analysensysteme GmbH, Langenselbold, Germany) via heat combustion at 1150 °C and thermal conductivity detection. The stable isotope ratio ^15^N/^14^N was measured as described in Franzaring, Gensheimer, Weller, Schmid and Fangmeier [20] using an isotope ratio mass spectrometer (Deltaplus XL, Thermo Finnigan, Bremen, Germany) connected by an open split device (ConFlow II, Thermo Finnigan, Bremen, Germany). The ^14^N/^15^N ratio was expressed relative to the isotopic signature of N_2_ in the air as δ values (in ‰).

### 2.9. Catalase Zymograms

Leaf discs (1 cm diameter) were taken from approx. the same position within the respective leaf and frozen in liquid nitrogen. After homogenization on ice with extraction buffer (100 mM Tris, 20% glycerol (*v*/*v*), 30 mM DTT, pH 8.0) samples were centrifuged for 15 min at 4 °C and 14,000 rpm. Total protein concentration of the supernatant was determined and 15 µg protein were separated via native PAGE (7.5% polyacrylamide, 1.5 M Tris, pH 8.8; running buffer: 25 mM Tris, 250 mM Glycin, pH 8.3). After protein separation, gels were rinsed twice with distilled water, incubated for 2 min in 0.01% hydrogen peroxide solution and then rinsed again twice with water. Incubation in staining solution (1% FeCl_3_ and 1% K_3_[Fe(CN)_6_]) was carried out under constant agitation until bands became visible (4–6 min). To stop any staining reaction, the solution was removed and the gel was rinsed with water.

## 3. Results

### 3.1. Transcriptome Analysis of OSR during Induction of Senescence under Two Different N-Supplies

To investigate the process of senescence induction and N-remobilization in OSR in more detail, microarray data made available by Safavi-Rizi et al. [19] were screened for genes related to nitrogen metabolism and nitrogen storage differentially expressed under optimal (N_O_) and low (N_L_) nitrogen. Among the genes that we identified to be related to nitrogen metabolism, we found a substantial number of genes encoding SSPs or proteins related to them, which were initially described to be exclusively expressed in seed tissue. However, here we found a reasonable expression in leaf tissue. An AGI code list with SSP and SSP-associated transcripts was compiled (Appendix A). Out of the 124 AGI identifiers, 17 *Bn* unigenes or gene clusters were differentially expressed in OSR according to development and/or N-supply. These could be assigned to 9 different expression profiles which have been defined by Safavi-Rizi and co-workers [19]; all colored profiles contain a significant number of temporally and N-dependently regulated genes [19]. Seven SSP and/or related genes were differentially expressed in leaf No. 4 and leaf No. 8 under both N-supplies, 15 under optimal N-supply (N_O_) and 14 under low N-supply (N_L_). A fairly high number of SSP related genes were differentially regulated in both leaves (11 out of 17), indicating that these transcripts appear to play a role during senescence progression at different canopies of the OSR plants (Figure 1B,C).

One important signaling molecule governing senescence regulation is hydrogen peroxide. Therefore, intracellular hydrogen peroxide contents of the same plant material were measured in leaf No.5 and already published in Bieker et al. [11]. Under optimal N-supply H_2_O_2_ reached its maximum levels at 12 weeks after sowing (85 DAS) and these high levels were maintained until week 13, while under low N the maximum was higher and was reached at week 13 (92 DAS). H_2_O_2_ declined at week 14 under both treatments. In order to get some hints whether H_2_O_2_ is involved in regulation of N-dependent senescence-associated gene expression and in the transcriptional regulation of the SSP-associated genes, we analyzed the correlation between H_2_O_2_ profiles and gene expression. Transcripts similar to *CRUCIFERIN3* (*CRU3*, At4G28520) and *MAIGO2* (*MAG2*, At3G47700) were induced in parallel to the H_2_O_2_ increase. CRU3 belongs to the SSPs of the Cruciferin superfamily whereas the MAG2 protein is involved in the ER exit of SSPs [30,31]. Interestingly, *SEED STORAGE ALBUMIN4* (*SESA4*, At4g27170) expression was reduced with rising H_2_O_2_ contents and induced when H_2_O_2_ levels declined again. This antagonistic behavior was also observed for *ECTOPIC EXPRESSION OF SSP1* (*ESSP1,* At2g19560) and several bifunctional-lipid transfer proteins (*BI-LTPs*). The same analysis was conducted for N_L_ data. Again, some SSPs were found to follow the H_2_O_2_ profile. However, instead of *CRUCIFERIN* transcripts, *NAPIN* (*SESA4*) peaked in its expression together with highest H_2_O_2_ levels, thus showing the exact adverse behavior compared to N_O_. Moreover, under N_L_ conditions, *MAG2* was found to be repressed with increasing H_2_O_2_ contents and expression increased again with decreasing H_2_O_2_ levels. Disregarding H_2_O_2_ contents, overall up-regulated transcripts included transcripts similar to *BRAHMA* (*BRM*, At2gG46020) and SSP processing enzymes (homologues to At4g32940 e.g., *GAMMA-VPE*, γ-vacuolar processing enzyme). Interestingly, the BRM protein mediates SSP repression in vegetative tissues. Down-regulated transcripts contained multiple homologues to bifunctional-lipid transfer proteins (*BI-LTPs*), *VACUOLAR SORTING RECEPTOR HOMOLOG1* (*VSR1*, At3G52850) and *ESSP1.*

### 3.2. Verification of SSP Expression via qRT-PCR and Western-Blot

In order to confirm the expression patterns of the mayor seed storage proteins, the 12S globulins (Cruciferins) and 2S albumins (Napins) via qRT-PCR and to verify the production of the corresponding proteins in leaves, another series of OSR plants was grown under constant greenhouse conditions. Again, an increase in SSP gene expression upon accumulation of H_2_O_2_ in *B. napus* leaves was observed. Moreover, timing of induction and repression of *CRUCIFERIN* coincided with the accumulation and decrease of intracellular H_2_O_2_. However, in contrast to the microarray-experiment, in this experiment both, *CRUCIFERIN* and *NAPIN* were up-regulated which might be due to different growth conditions and uncontrolled N-supply. In addition, protein levels were analyzed via Western blot and subsequent immune detection of SSPs. Here, a progressive accumulation pattern emerged, starting with SSP production in the leaves at the lower canopy level when plants were still young. Later, expression continued with a build-up in upper canopy leaves when development progressed and SSPs usually start to accumulate in siliques and seeds. In addition, accumulation of SSPs was also observed in shoot tissue (Figure 2).

To confirm the protein expression also under controlled N conditions as described in references [19,20], leaf positions 4, 8 and 12, the terminal leaf as well as shoot tissue and siliques were harvested of another growth series. Chlorophyll contents were measured via *atLeaf+*. Leaf No.4, the lowest leaf position analyzed, remained on the plant only until week 8 under N_O_ conditions, whereas under N_L_ conditions this leaf was shed between week 11 and 12. Longer maintenance of photosynthetic capacities, especially of lower canopy leaves, enhanced assimilate supply to the pods as well as the roots [22]. The higher canopy leaves exhibited a slightly higher Chl-content under higher N-supply.

Under N_O_ conditions, quantification of detected CRU proteins showed a similar pattern to the *CRU* transcripts in the microarray and the qRT-PCR experiments. A stepwise increase in CRU protein contents in the leaves was observed beginning at the lowest positions with peak levels between weeks 8 and 10 (Figure 3A). Under N_L_ conditions CRU production was reduced approx. 10-fold and no comparable pattern became clear. For NAPIN, the detected amount of protein was much higher under N_L_ conditions than under N_O_ like in the microarray expression data. Accumulation patterns were similar, but expression was more pronounced in the lower leaf positions, analogous to the prolonged Chl retention observed there. However, a direct comparison between NAPIN and CRUCIFERIN contents is not possible as both proteins were detected with different antisera. Therefore, the signals were normalized to the respective signals of the controls. Nevertheless, both proteins could be detected not only in leaves but also in stems (Figure 3A,B,D). Moreover, total N-content in the seeds was approx. 25% higher under N_O_ than under N_L_ conditions (Appendix A). Gironde et al. [32] have already indicated an interim storage function for shoot tissue upon asynchronous senescence and seed filling. Our results support this possible function. Even under N-starvation conditions, where a reduction of H_2_O_2_ contents and barely detectable CRU synthesis was shown in leaves, still a minimal build-up of SSPs occurred in the shoot (Appendix A).

The idea of an interim N-storage is also supported by the follow-up of ^15^N-labeled nitrogen over time in parallel to the expression profiling [21]. ^15^N-labeled fertilizer was supplied at three different time points, directly after sowing (DC 0), 72 DAS (DC 30), and 80 DAS (DC 59). Nitrogen recovery and the distribution of the labeled N to different organs was measured by isotope ratio mass spectrometry [21]. Here, we present the δ^15^N mean values for pooled green and senescent leaves as well as for the individual leaves No. 4, 5 and 8 at harvest 4 (92 DAS). Exactly the same leaf material was also used for expression profiling (No. 4 and 8; [19]) and hydrogen peroxide measurements (No. 5; [11]). At this time point, early supplied nitrogen (DC 0) appeared to be inefficiently remobilized and mainly rested in the older leaves (No. 4 and 5) while the highest proportion was shed in senescent leaves. In contrast, nitrogen originating from the second and third N gift (DC 30 and DC 59) accumulated less in shed senescent leaves but more in the younger leaf No.8. This can be observed under both N-supplies in which the effect is most prominent under N_L_ and DC 30 labelling (Figure 4). This might indicate that N taken up at flowering time appears to be stored temporarily in the younger leaves most likely to be further remobilized to the developing seeds after anthesis. Possibly, also stems and pods contribute to this interim storage, as total N in stem and pods declined while total N in seeds increased (Appendix A).

Taken together, our results indicate that a transcriptional control of SSP expression via H_2_O_2_ but also via the plants’ N-status appears to be in place. Therefore, we wanted to identify possible regulatory elements (RE) in the promoter regions of the respective genes by in silico analyses. As *Brassica napus* was formed ~7500 years ago by hybridization between *B. rapa* and *B. oleracea*, followed by chromosome doubling, and most orthologous gene pairs in *B. rapa* and *B. oleracea* remain as homologous pairs in *B. napus* [33], we analyzed the promoter regions of both SSP gene copies, respectively. The NSITEM-PL program with RegSite PL Database of Plant Regulatory Elements (Release 14, May 03, 2014; default parameters were set) [34] tests for the presence of known REs in conjunction with positional conservation within the supplied sequence sets comprising 3kb upstream sequences of *B. oleracea* and *B. rapa* as well as of the *B. napus CRUCIFERIN* or *NAPIN* homologues. Remarkably, several binding sites for transcription factors involved in either ROS response or N-management or both were identified (Appendix A). For example, binding elements for TGACG-BINDING FACTOR2 (TGA2, a bZIP transcription factor involved in general ROS-and pathogen-response, TEOSINTE BRANCHED1/CYCLOIDEA /PROLIFERATING CELL FACTOR20 (TCP20, involved in systemic N-status signaling and leaf-senescence) and ELONGATED HYPOCOTYL5 (HY5, a ROS-responsive bZIP transcription factor involved in N management) were found (for a complete listing see Appendix A).

### 3.3. Hydrogen Peroxide Signaling under Complete N-Starvation

To explore a possible interplay between the plants’ nitrogen status and H_2_O_2_ as signaling molecule during the induction of leaf senescence and SSP expression, N-starvation experiments with OSR plants grown on hydroponics were conducted. As expected, N-starvation led to the induction of premature senescence indicated by the decrease in chlorophyll contents, especially in older leaves. However, no increase in intracellular H_2_O_2_ contents could be measured, as would have been expected for a senescence signaling molecule, but rather a slight reduction of H_2_O_2_ was determined (Figure 5). Nevertheless, a reduction of catalase activity was observed (Appendix A) as was already described during developmental senescence under full N-supply [11].

Arabidopsis plants subjected to N-starvation on hydroponic culture consistently exhibited similar effects. Shortly before anticipated senescence induction, hydroponic medium was switched from full nutrient supply to N-free medium. Now, an even more pronounced reduction of H_2_O_2_ contents was detected (Figure 6). However, in both cases, an extensive production of anthocyanins became visible. In Arabidopsis, anthocyanin production was strongest at vasculature during the early phases of N-depletion (Figure 6D). In OSR, anthocyanin accumulation was also observed after N-depletion (Figure 5C,D) but for OSR the vasculature seemed to remain with very low anthocyanin production (Figure 5D). Obviously, OSR leaf vein and leaf laminae are tissues with different regulatory mechanisms during senescence. Besides other functions, anthocyanins were discussed to act as ROS scavenging molecules. Accordingly, reduction of intracellular H_2_O_2_ was observed in N-starved Arabidopsis as well as in OSR leaves (Figure 5A,B and Figure 6A,B). For both species, Chl-contents remained constant in young leaves during the treatment, while older leaves showed a significant reduction in Chl-contents (Figure 5A,B and Figure 6A,B). These findings tempted us to speculate that one possible role of anthocyanin production during N-starvation-induced senescence is the scavenging of the signaling molecule H_2_O_2_ and by that a suppression of at least part of the developmental senescence program. SSP accumulation under complete N-starvation conditions was barely detectable in leaf tissue (Appendix A).

## 4. Discussions

Genome-wide expression analyses of OSR grown under two different N-regimes revealed that, according to the gene expression profiles, long-term and mild N-deficiency provokes accelerated senescence in lower canopy leaves but delayed senescence in upper canopy leaves [19]. The most important feature of senescence is the remobilization of nutrients out of the senescing tissue into developing parts of the plants, especially to fruits and seeds. By ^15^N-labelling at different time points in the same experiment, we could show that N taken up early during development was not efficiently remobilized and a high proportion of the labeled N was shed with senescent leaves. N taken up at flowering time appeared to be better remobilized and stored temporarily in the younger leaves. It was also most likely to be further translocated to the developing seeds after anthesis. This effect is most prominent under low N-supply (Figure 4) when N-remobilization is most likely to be more important. N-recovery in general and to the seeds was improved under low N, as 50.7% of the total N-gift was deposited in the seeds compared to 44.5% under N_O_ conditions [21]. A genome-wide expression analyses via microarrays of exactly the same plant material offered the possibility to analyze the expression profiles of genes involved in N-remobilization and N-storage (Appendix A). The most striking result for us in this analysis was that transcripts of seed storage protein genes or genes related to them, which were formerly thought to be restricted to seed tissue, were identified in leaves at different canopies like leaf No. 4 and No. 8 (Figure 1). Moreover, not only the transcripts but also the proteins were detected in leaf tissue (Figure 2 and Figure 3). However, the accumulation of these storage proteins was not equally distributed across all leaves but a wave-like spreading of SSPs was observed starting in the oldest leaves when plants were still young and climbing up into younger leaves with the progression of development and declining again in later stages with a residual SSP content in the terminal leaf (Figure 2). SSP protein accumulation was also detected in shoots and pods (Figure 3). Considerable differences were observed between the two different N regimes: While *NAPINS* appeared to be more highly expressed under N_L_ conditions, more *CRUCIFERINS* were produced under higher N supply, creating the SSPs possible candidates for interim storage of N in leaves during N remobilization processes of senescence. As CRUCIFERINS and NAPINS have distinct molecular characteristics and functionalities [4], it makes sense that these two classes of SSPs are also expressed in contrary ways under different nitrogen regimes in leaves. SSP proteins also detected in stems and pods and total N-analyses in stems and pods additionally points to interim N-storage in these tissues as total N in stem and pods declined while total N in seeds increased (Appendix A). In maize *opaque-2* knock-out plants, storage protein synthesis significantly reduced the pool of free AAs in mature endosperm [35] and could thus enhance sink strength for further N-remobilization. A similar mechanism could possibly operate in leaf tissue as source strength of the disintegrating leaves might exceed sink strength of the newly developing organs [32]. Therefore, the pool of free AAs could be reduced by the built-up of storage proteins decreasing the source strength of the respective leaves. As such, a stepwise N remobilization from lower leaf canopies could take place, which coincides with the wave-like expression of the SSPs (Figure 2). How this is coordinated in oilseed rape plants is still an open question. However, accurate QTL mapping and potential candidates identified based on high-density linkage map and BSA analyses revealed that complex genetic mechanisms for oil and protein accumulation in the seeds of rapeseed are in place [6]. Further analyses of these candidate genes will allow for finding out whether they are also expressed in leaf tissue and contribute to storage reserves in seeds through an interim storage of N via SSPs in leaves. To function as an efficient interim storage, two main prerequisites need to be fulfilled, namely an efficient and compact storage and an easy access for remobilization [32], which both holds true for SSPs. SSPs are usually stored in higher complexes with multimeric conformation in protein storage vacuoles [5]. To achieve this conformation, extensive processing of pre-cursor proteins by VPEs (vacuolar processing enzymes) is necessary. Indeed, γ-VPE (AT4G32940) and other VPEs were differentially expressed in leaves under both N treatments (Appendix A) indicating that SSP storage in a processed multimeric form is also possible in leaves. In the three plant growth series conducted under different conditions, *CRUCIFERIN* and *NAPIN* expression and accumulation correlated with intracellular hydrogen peroxide contents (Figure 2, [11]), which might be the link to the regulation of senescence processes, as hydrogen peroxide is a well-known signaling molecule in leaf senescence. Many transcriptional regulators and SAGs are up-regulated by increasing intracellular hydrogen peroxide concentrations. In silico analyses of 3 kbp upstream of the coding regions of *CRUCIFERIN* and *NAPIN* genes also supported the existence of a direct link between SSP expression and oxidative status of the cells (Appendix A). Spatiotemporal occurrence and distribution of SSPs and the corresponding mRNAs were shown to be tightly regulated and were so far thought to be restricted to the developing seeds [36]. In consistence, most of the *cis* elements identified by *in silco* analyses of the SSP promoter regions (Appendix A, Appendix A) are characteristic for seed-specific expression in various species. Multiple abscisic acid (ABA) responsive elements were found in *CRUCIFERIN* as well as *NAPIN* upstream regions. ABA is known to be a key factor during seed maturation by suppressing premature germination and regulating a variety of processes during embryogenesis [37,38]. This also points to seed-specific expression. During SSP deposition, ABA contents steadily rise until they decrease again during desiccation. However, ABA is also a senescence-promoting hormone accumulating during developmental as well as N-starvation induced senescence [39,40]. For *CRUCIFERINs*, besides the known RY/G-Box and ABA responsive elements, a further motif for seed specific *GLUTELIN* expression known from *Oryza sativa* (OSMYB5 binding site) was found [41]. Additionally, an OPAQUE-2 binding site, a *cis* element identified in *Zea mays* regulating *ZEIN* expression [42], was found to be present in *CRUCIFERIN* upstream sequences.

Studies on SSPs in *A. thaliana* have shown the interplay of the bZIP transcription factors bZIP53, bZIP10 and ABI3 to be necessary for full promoter activation [43]. Simultaneous overexpression of all three factors led to most pronounced promoter activation. Although expression of one of these factors alone showed only marginal *SSP* promoter induction in transient assays, the authors have depicted the crucial role of bZIP53, since constitutive *bZIP53* overexpression *in planta* was able to induce seed-specific promoter activation in non-seed tissue [43]. Therefore, we screened the OSR microarray data for the expression of the corresponding transcription factors. None of these transcription factors (TFs) were expressed in leaves throughout development of the OSR plants and transient overexpression of *AtbZIP53* together with a *BnSSP:GUS* construct in Arabidopsis protoplasts did not change expression of the reporter gene (Data have been provided as non-published material). Accordingly, a different mechanism of leaf-specific expression has to be in place in OSR.

The overall SSP expression pattern suggests a correlation with the developmental senescence program coupled to ROS contents and N availability. In 2001, Desikan and colleagues already identified Arabidopsis seed storage proteins to be responsive to oxidative stress [44]. Here, expression of *NAPIN* and *CRUCIFERIN* coincided with the hydrogen peroxide peak in the microarray experiment and was also confirmed by qRT-PCR in a second growth experiment on soil (Figure 2). Moreover, this also concurs with our database-aided motif identification. *Cis* elements known for redox-responsiveness were found in the upstream sequences of both SSP-classes. Besides the direct ROS responsive elements of *HY5*, other motifs suggesting an indirect coupling to ROS were identified. The LS7 Box of the PATHOGENESIS RELATED GENE1 (PR-1) and the G-Box of the *VEGETATIVE STORAGE PROTEIN-A* gene (*VSP-A*) of soybean, which are both involved in pathogen responses and ROS bursts, were identified [45,46,47]. Furthermore, we found a TGA2 binding site. TGA2 is involved in salicylic acid and late jasmonic acid signaling pathways, and is an interaction partner of NON-EXPRESSER OF PR-GENES1 (NPR1), which acts as a central transcription activator of many defense-related genes. NPR1 is redox sensitive and cytoplasmic H_2_O_2_ prevents the translocation of NPR1 to the nucleus [48]. Earlier publications have reported that SSPs can be massively oxidized, especially by carbonylation [49,50]. Besides an easier access to SSP monomers after multimer destabilization upon carbonylation, a ROS scavenging function has been proposed [51]. Recently, Nguyen et al. [52] have shown anti-oxidative functions for CRUCIFERINs in Arabidopsis seeds. Seeds generated from Arabidopsis triple *cruciferin* KO lines (*crua/crub/cruc*) displayed a much higher sensitivity to artificial ageing as well as considerably higher protein carbonylation.

More evidence for N-status dependent transcription of SSPs is given by the presence of a TCP20 binding site in *NAPIN* upstream regions. TCP20 participates in systemic N-status signaling as well as in leaf senescence [53,54]. This dependency on the plant N-status is also reflected in our analyses of the microarray expression data. When plants were grown under N_L_ conditions, the expression of *NAPIN* was induced whereas under N_O_ conditions, *CRUCIFERIN* was predominantly expressed. Thus, the type of SSP which is expressed in leaf tissue under different N availability might be selected by its capability to store N as the amount of N per protein molecule is more than 3-fold higher for CRUCIFERIN than for NAPIN. CRUCIFERIN (488 amino acids) is much larger than NAPIN (159 amino acids) and thus requires more input but also has more N storage capacity (698 N atoms vs. 228 N atoms, respectively). A similar mechanism for nutrient availability-dependent SSP expression was shown already for sulfur. Low sulfur supply increased the ratio between S-rich CRUCIFERIN and S-poor NAPIN significantly and reduced total fatty-acid contents [1]. Accordingly, SSP accumulation under complete N-starvation conditions was barely detectable in leaf tissue, as an interim storage is not necessary under these conditions. Remarkably, also no H_2_O_2_ accumulation was observed (Figure 5, Appendix A). However, how differential expression of *CRUCIFERIN* under N_O_ and *NAPIN* under N_L_ is achieved and which role hydrogen peroxide exactly plays, still needs to be further investigated. Besides TCP20, HY5 is also not solely involved in ROS responses but also in N management by regulating expression of *NITRITE-REDUCTASE-1* (*NIR1*) and *AMMONIUM-TRANSPORTER-1;2* (*AMT1;2*) in dependence of N-supply [55]. Moreover, HY5 can interact with PHYTOCHROME B (PHYB) during prolonged red-light exposure and induce EDS1 (*ENHANCED DISEASE SUSCEPTIBILITY-1*), which then promotes ROS production [56]. This interaction hints at a possible link to PHYTOCHROME-INTERACTING FACTOR7 (PIF7). PIF7 binding elements are present in *NAPIN* upstream sequences and could be targeted by PHYB-recruited PIF7 as part of a HY5-PHYB response.

Another indirect mechanism might be emanating from a NAC (NAM/ATAF1/2/CUC2) recognition site (NACRS) in *NAPIN* upstream regions, which is known to be bound by ANAC019, 055 and 072 [57]. Thus, another link to the senescence program in general is given. These factors take part in the regulation of major chlorophyll catabolic genes and ANAC019 has been identified as an activator of senescence in Arabiodpsis [58,59]. More recently, a *Brassica napus* NAC factor, NAC87, which also binds the NACRS, has been characterized to regulate both, ROS generation and leaf senescence regulatory genes [60]. An additional indirect N-dependent regulation might be exerted by GNC (**G**ATA, **N**itrate-inducible, **C**arbon metabolism-involved), which controls expression of ANAC019 [61].

Complete N-starvation appeared to elicit premature senescence and most likely a stress-induced N-remobilization program. A general indicator for several stress responses is the induction of the phenylpropanoid pathway, which in turn leads to lignification, anthocyanin production and build-up of other repellents and protective secondary metabolites. Several studies in different plant species have shown the induction of the phenylpropanoid pathway or an increase in anthocyanin production provoked by N-limitation [62,63,64]. In our experiments, anthocyanins were present in high amounts, so that they were visible with the naked eye and most likely were able to reduce ROS contents. This might explain why despite an induction of premature senescence and reduction of catalase activity no increase of H_2_O_2_ was measured (Figure 5 and Figure 6, Appendix A). In contrast, H_2_O_2_ levels were even reduced. The hypothesis that anthocyanins scavenge substantial amounts of H_2_O_2_ and thus simply mask the increment in ROS contents will be further investigated by experiments with anthocyanin synthesis mutants *transparent testa* (*tt*) in Arabidopsis. Remarkably, *tt* mutants grown under limiting nitrogen supply did not show a difference in senescence measured by chlorophyll loss and *SAG12* expression [65]. Moreover, non-enzymatic scavengers like ascorbate and tocopherol are elevated during senescence under low nitrogen in OSR in the leaf laminae as well as in the leaf veins [66]. However, *nla* (*nitrogen limitation adaptation*) mutant Arabidopsis plants grown under low N could not accumulate anthocyanins and instead produced a severe N-limitation-induced early senescence phenotype, perhaps due to the fact that the senescence-inducing H_2_O_2_ signal was not quenched by anthocyanins [63]. Therefore, anthocyanin production under N-starvation might also serve the repression of at least the H_2_O_2_-controlled part of the senescence program. While reduction of catalase activity hints at a promotion of the standard developmental senescence program (Appendix A), the apparent decrease in H_2_O_2_ contents contradicts this assumption (Figure 6). In accordance with these observations, comparative expression analyses have shown strong differences between stress-induced and developmental senescence programs in early stages, which then converged in later stages [67]. Taken together, a complex interplay between intracellular hydrogen peroxide contents and nitrogen availability appears to govern the senescence program and N-remobilization efficiency.

## Figures and Tables

**Figure 1 genes-10-00072-f001:**
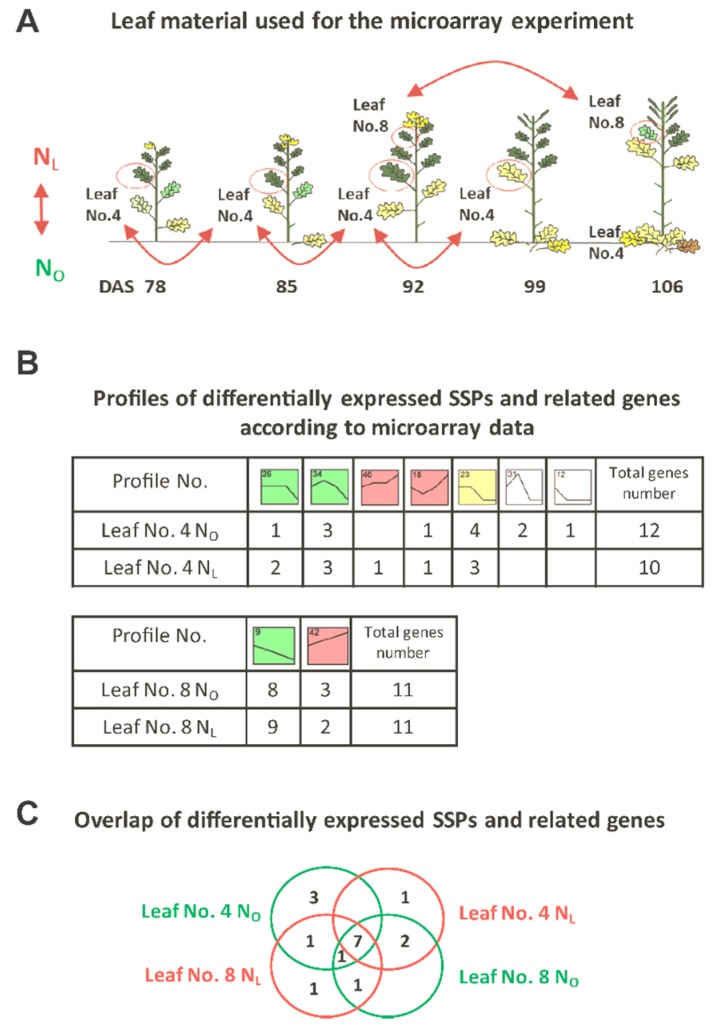
SSPs and related genes differentially expressed during development and/or N-supply assigned to 9 different expression profiles. (**A**) Illustration of the leaf material used in the microarray experiment (modified after Safavi-Rizi et al. [19]). Leaf No. 4 was analyzed at 4 different time points, leaf No.8 was analyzed at 2 different time points under N_L_ or N_O_ supply, respectively. (**B**) Number of SSPs and related genes, which were differentially expressed in leaf No. 4 and leaf No. 8 under N_L_ and under N_O_ conditions. Each box represents one of 50 pre-defined expression profiles [19]. The number of each model expression profile is shown in the top left corner of each box, all colored profiles contain a significant number of temporally and N-dependently regulated genes. Profiles with the same color are very similar and defined as one cluster in Safavi-Rizi et al. [19]. (**C**) Overlap of differentially expressed SSPs and related genes between leaf No. 4 and No. 8 and the growth conditions N_L_ and N_O_.

**Figure 2 genes-10-00072-f002:**
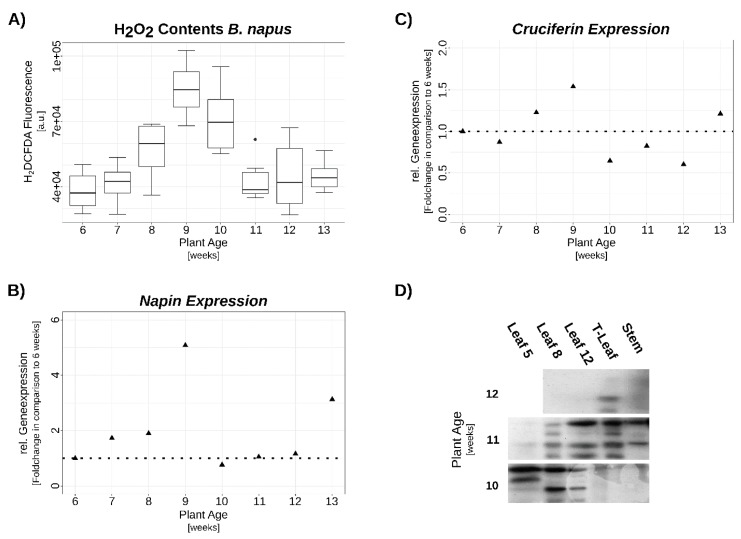
OSR developmental series. Exemplary plants grown in the green house under stable conditions are shown at the top, numbers indicate weeks after sowing. (**A**) H_2_O_2_ contents in leaf No. 8. (**B**) and (**C**) relative gene-expression of *NAPIN* and *CRUCIFERIN*, respectively, analyzed by qRT-PCR. Expression was normalized to *ACTIN2* and expressed relative to value at 6 weeks. (**D**) Western-blot of the corresponding plant material immune-detected with anti-NAPIN antiserum. Analyzed proteins are samples of 3–5 biological replicates pooled in equal amounts. Leaves are numbered according to their appearance; terminal leaf (T-leaf) is the leaf residing directly below the first developing pods. H_2_O_2_ data are medians +/−1.5xIQR of at least 3 biological and 2 technical replicates each. Gene expression data are means of 3 technical replicates from pools of at least 3 biological replicates.

**Figure 3 genes-10-00072-f003:**
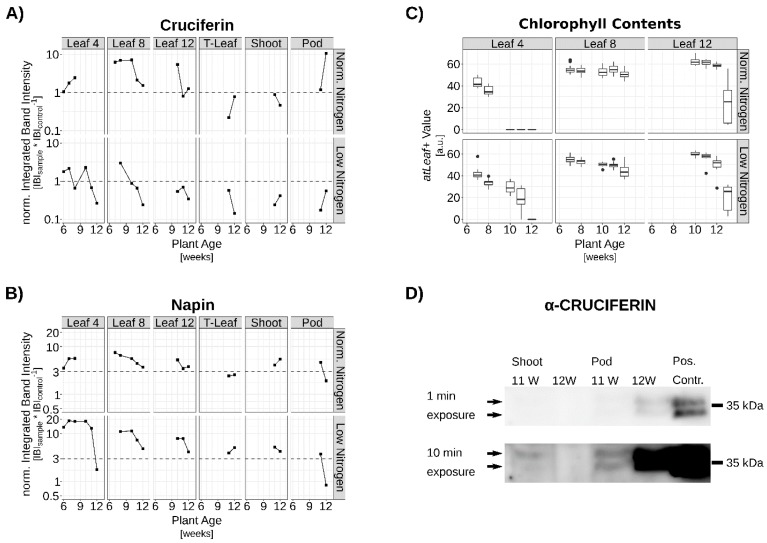
Seed storage protein (SSP) levels of OSR plants grown under normal and limiting nitrogen (N) supply. Quantification of (**A**) CRUCIFERIN and (**B**) NAPIN signals. Material analyzed are samples of 3–5 biological replicates pooled in equal amounts. (**C**) Corresponding chlorophyll contents (median +/−1.5xIQR of 3 biological replicates). Upper graphs normal N-supply, lower graphs N-limitation. (**D**) Exemplary signal of stem and pod material detected in a Western blot (marked by arrows, 25 µg total protein). One min exposure (top) to show non-oversaturated positive control (15 µg total protein) and 10 min exposure for the shoot and pod signals (bottom).

**Figure 4 genes-10-00072-f004:**
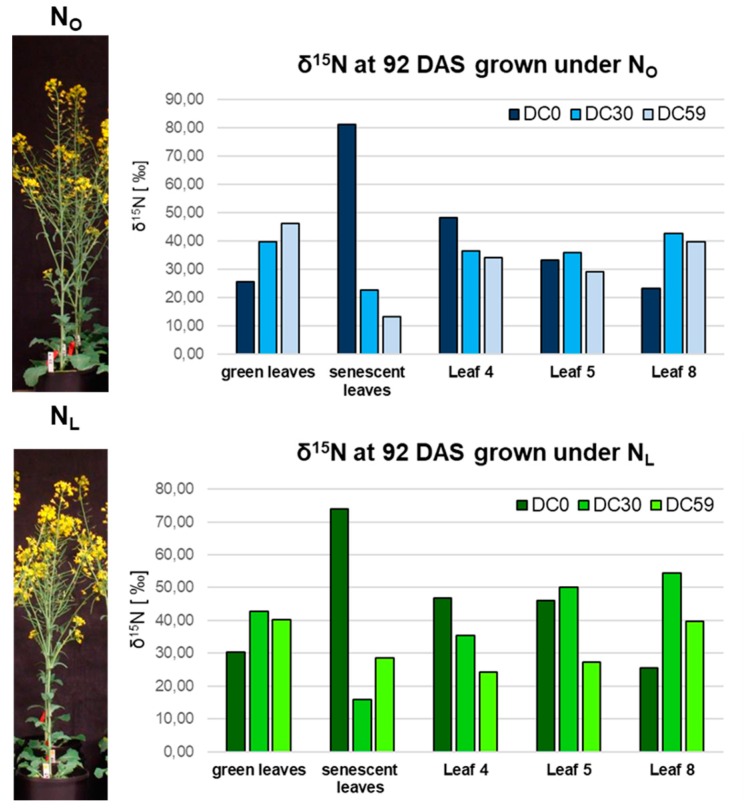
Nitrogen recovery and the distribution of the labeled N to different organs. δ^15^N signatures at 92 DAS (days after sowing) in pooled green and senescent leaves as well as in leaves No. 4, No. 5 and No. 8 of plants grown under N_L_ (lower chart) and under N_O_ (upper chart) conditions are presented (DC0: fertilization at 0 DAS, “old N”, DC30: fertilization at 72 DAS, DC59: fertilization at 80 DAS, “new N”). The stable isotope ratio ^15^N/^14^N was related to the isotopic signature of N_2_ in the air and is reported here as “delta” δ values in ‰. Exactly the same plant material was used for the microarray analyses of Safavi-Rizi et al. [19] and physiological analyses in Franzaring et al. [20,21].

**Figure 5 genes-10-00072-f005:**
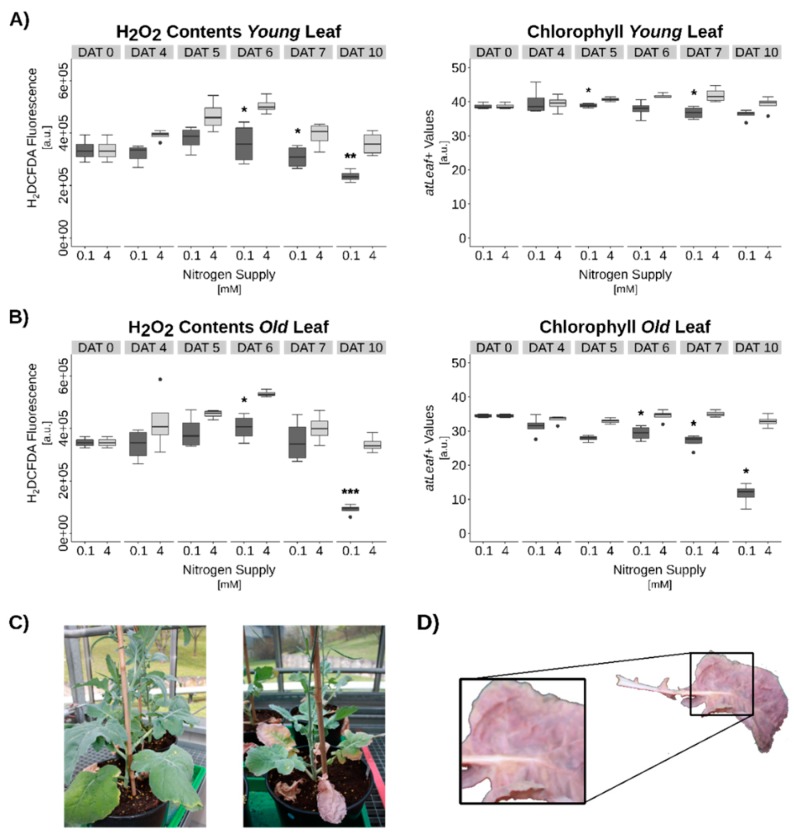
Nitrogen (N) starvation induced senescence in *Brassica napus*. (**A**) and (**B**) Hydrogen peroxide (H_2_O_2_) and chlorophyll contents in young leaves (**A**) and in old leaves (**B**). (**C**) Plant grown under full N-supply (left) and under N-starvation (right). (**D**) Exemplary anthocyanin accumulation pattern in N-starved plants. Statistics: Welch’s t-test. Significance levels: *p* < 0.001 = ***; *p* < 0.01 = **; *p* < 0.05 = *. Data are medians +/−1.5xIQR of at least 3 biological and 2 technical replicates each.

**Figure 6 genes-10-00072-f006:**
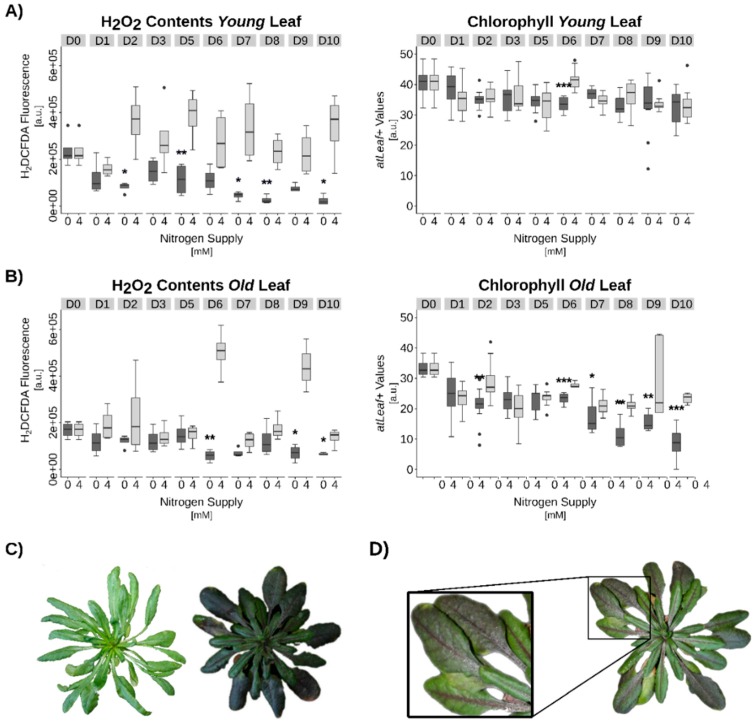
Nitrogen (N) starvation induced senescence in *Arabidopsis thaliana*. (**A**) Hydrogen peroxide (H_2_O_2_) and chlorophyll contents in young leaves, (**B**) in old leaves. (**C**) Left: plants grown under full N-supply, right under N-starvation. (**D**) Exemplary anthocyanin accumulation pattern in N-starved plants. Statistics: Welch’s t-test. Significance levels: *p* < 0.001 = ***; *p* < 0.01 = **; *p* < 0.05 = *. Data are medians +/−1.5xIQR of at least 3 biological and 2 technical replicates each.

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
