# Peer review of "Nitrogen Supply Drives Senescence-Related Seed Storage Protein Expression in Rapeseed Leaves"

_genes, 2019, doi:10.3390/genes10020072_

Reviewer 1 Report

Comments

In this manuscript authors reported the major classes of seed storage proteins to be expressed in vegetative tissue and expression of SSPs was correlate with intracellular H2O2 contents, which functions as well-known signaling molecule in developmental senescence.

 There are some other issues that need to be resolved, before its final acceptance.

 1. Title of manuscript is too long, should be reduced.

 2. The introduction provides a good, generalized background of the topic that quickly gives an idea on background. However, to make the introduction more substantial, the author may wish to provide several recent related references (one recent review given below). Further, third paragraph of introduction “Developmental senescence ………stem and reproductive tissue” written in more details, I think this information already available in literature, so this para should be concise.

Gacek, K., Bartkowiak-Broda, I., & Batley, J. (2018). Genetic and Molecular Regulation of Seed Storage Proteins (SSPs) to Improve Protein Nutritional Value of Oilseed Rape (Brassica napus L.) Seeds. Frontiers in plant science, 9, 890. doi:10.3389/fpls.2018.00890

3. In Result section, Line 225-231 (To investigate the…..figure 1A); Line 288-291(another series…..were sampled); Line 313-316 (As these western ……and Fangmeier); Line 370-375 (N15 signature…. and Fangmeier) Line 399-404 (OSR plant…..control plants) these paragraphs were belongs to Martial and method and should be shifted there.

 4. In Figure 2, first plate on top showing OSR plants grown in the green house under stable conditions are not required in my view. Secondly please mention age of plants instead triangular representation in part D of Figure 2 showing Western-blot.

 5. Discussion section is not connected to results. So, I suggested that authors should improve the discussion section. Whole discussion mainly focuses on recognition sites in NAPIN/ CRUCIFERIN upstream regions. They have to discuss their results and comparison with earlier and recent published papers in more depth. 

Author Response

Response to the reviewer’s comments:

Reviewer 1:

We have addressed the single point, which was made by reviewer 1, and have directly mentioned the concentration of nitrogen applied to the plants for all experiments.

Reviewer 2 Report

General:

Overall I think this paper is just great. It's written really well, clear, and covers all questions that pop into my head as I am reading through the results. So much data has gone into this work, and the authors should be pleased at the quality of research they have produced. I only provide very minor comments/questions in my review.

Abstract: great

Introduction: Well written and clear. If I had to pick on something, I would suggest just a couple lines introducing the concept of H2O2 in senescence, just as it is mentioned in a few places throughout the manuscript.

Methods:

Section 2.1

Can the authors please define how much nitrogen was fed to the soil during the experiments? (i.e Nitrogen content of fertiliser etc). I do realise the authors quote papers for nitrogen supply, however for ease of reading (and clarity) it would be helpful to have the ppm of nitrogen readily available. 

Methods are otherwise nicely written.

Results:

The results are presented well, and are clearly outlaid. Nice.

Discussion:

The discussion flows well and covers all aspects and questions I had as a reviewer after looking at the data, great.

Author Response

Response to the reviewer’s comments:

Reviewer 2:

We have address all points made by reviewer 2:

1.) The title has been changed and was shortened.

2.) We included the suggested review and several recent publications in the introduction and shortened the third paragraph.

3.) The suggested paragraphs have been moved to the material and method section.

4.) Figure 2 has been changed according to the reviewer’s suggestions

5.) As the other reviewer was very happy with the discussion, we did not change the overall structure but included some new aspects.

Round  2

Reviewer 1 Report

All corrections were made in the manuscript and manuscript may be accepted for publication.